**Data Availability Statement:** The de-identified data for all the OGTTs analysed in this study are in a

# Development of a fasting blood glucose-based strategy to diagnose women with gestational diabetes mellitus at increased risk of adverse outcomes in a COVID-19 environment

Michael d'Emden[1,2]*, Donald McLeod[3], Jacobus Ungerer[4], Charles Appleton[5], David Kanowski[6]

1 Royal Brisbane and Women's Hospital, Herston, Queensland, Australia, 2 University of Queensland, Brisbane, Australia, 3 Department of Endocrinology and Diabetes, Royal Brisbane and Women's Hospital, Herston, Queensland, Australia, 4 Pathology Queensland, Herston, Queensland, Australia, 5 Queensland Medical Laboratories, Murrarie, Queensland, Australia, 6 Sullivan Nicolaides Pathology, Indooroopilly, Queensland, Australia

* Michael.d'Emden@health.qld.gov.au

## Abstract

### Objective

To evaluate the role of fasting blood glucose (FBG) to minimise the use of the oral glucose tolerance test in pregnancy (POGTT) for the diagnosis of gestational diabetes mellitus (GDM).

### Research design and methods

We analysed the POGTTs of 26,242 pregnant women in Queensland, Australia, performed between 1 January 2015 and 30 June 2015. A receiver operator characteristics (ROC) assessment was undertaken to indicate the FBG level that most effectively identified women at low risk of an abnormal result.

### Results

There were 3,946 (15.0%) patients having GDM with 2,262 (8.6%) having FBG ≥ 5.1mmol/l. The ROC identified FBG levels >4.6mmol/l having the best specificity (77%) and sensitivity (54%) for elevated 1 and/or 2hr BGLs. There were 19,321 (73.7%) women having FBG < 4.7mmol/l with a prevalence of GDM of 4.0%, less than 1/3rd the overall rate. Only 4,638 (17.7%) women having FBGs from 4.7–5.0mmol/l would require further evaluation to confirm or exclude the diagnosis.

### Conclusion

This contemporary study of women across the state of Queensland, Australia suggests the FBG can be used effectively to define glucose tolerance in pregnancy, minimising their contact with pathology laboratories and potential exposure to the corona virus. This analysis,

single excel spread sheet and have been uploaded with this submission.

**Funding:** The author(s) received no specific funding for this work.

**Competing interests:** The authors have declared that no competing interests exist.

used in conjunction with outcome data from the HAPO study, provides reassurance to women and their health professionals that FBG < 4.7mmol/l has both a low rate of abnormal glucose tolerance and minimal adverse pregnancy-associated complications.

## Introduction

The new glucose criteria for the diagnosis of GDM were developed by the International Association of Diabetes in Pregnancy Studies Groups (IADPSG) based on the Hyperglycemia and Adverse Pregnancy Outcomes (HAPO) study [1, 2]. On this basis, the diagnosis of GDM is established on a POGTT if the fasting blood glucose level (FBG) $\geq$ 5.1 mmol/l, the 1-hr BGL $\geq$ 10.0 mmol/l, or 2-hrs BGL $\geq$ 8.5 mmol/l [2]. The pandemic of COVID-19 that has occurred across the world in 2020 has resulted in many changes in social behavior to minimise transmission of the virus. Social distancing is a corner stone of that strategy [3]. Performing an OGTT in the morning on a fasting pregnant woman creates a practical problem of potentially exposing women to a high risk environment for 2 to 3 hours at the busiest time of the day for a collection facility. Many women and health care practitioners are questioning the need to perform a POGTT. Minimising the time spent at the collection facility is consistent with strategies seeking to ensure adequate social distancing. Several peak bodies have developed alternative diagnostic strategies to minimize the need for a POGTT to diagnose GDM although potentially these may result in higher rates of pregnancy associated complications [4].

There are other practical problems. Many patients do not tolerate the POGTT, with significant rates of abdominal discomfort, nausea and vomiting invalidating the test [5, 6]. An increasing number of pregnant women have had bariatric surgery prior to becoming pregnant and an POGTT is not recommended in this situation due to its impact on gastric motility and absorption [7–9]. In consideration of these issues, we systematically evaluated the relationship of FBG to the 1 and 2-hr BGLs on a POGTT and their relationship to available published outcome data to identify a threshold FBG level associated with a reduced risk of abnormal glucose tolerance and minimal increase in pregnancy associated complications. This strategy could potentially negate the need to undertake for the POGTT for many patients, minimising their time spent in the higher risk environment of pathology laboratories.

## Methods

### Population and procedures

We evaluated the de-identified data of all patients having POGTTs (data available at 0, 1 and 2-hrs) at the three largest pathology laboratories (Queensland Medical Laboratory [QML], Sullivan Nicolaides Pathology [SNP], and Pathology Queensland (PathQ]) in the state of Queensland, Australia from January 1, 2015 to June 30, 2015. The tests were performed in hospitals or satellite collection centres in metropolitan, regional and remote locations throughout the whole state. Minimal baseline (date of test, length of gestation) and no clinical outcome or management data were available. The percentage of pregnant patients tested in Queensland during the study period could be estimated from the number of neonatal screening tests for hypothyroidism performed on every living child in Queensland after birth by PathQ, accepting that this may slightly over-estimate the number of pregnancies when there were multiple pregnancies.

All POGTTs followed a standard protocol in all three laboratories being conducted after an overnight fast with no recommendation to consume a high carbohydrate diet for 72 hours

prior to the test. Patients were seen between 7:00 and 9:00 am and were asked to ingest 75 g of glucose within 10 minutes. Patients remained in the collecting center resting quietly for the following two hours. The 0-hr time point refers to time prior to the commencement of the ingestion of glucose and the 1 and 2-hr time points refer to the time post commencement of ingestion of the glucose load. Collectively, these two BGL values are subsequently termed the post-load BGLs. Blood samples were centrifuged in serum separating tubes within 30 minutes as per laboratory protocols according to the manufacturers' recommendations, then stored at 4°C to minimise glycolysis. The length of gestation for SNP and PathQ (this information was not available from QML) was either written on the request slip by the referring doctor or volunteered by the patient; it was not verified by any other means. GDM was diagnosed according to the Australian Diabetes In Pregnancy Society (ADIPS) recommendations based on the IADPSG criteria [2, 10].

### Analysis of relationship of FBG to post-challenge BGLs

The relationship between FBG and post-load BGLs was assessed using linear regression, and we determined the point prevalence and the cumulative total of abnormal post-challenge BGLs at increasing FBG levels. We assessed the sensitivity and specificity of varying FBG cut-points from 4.0 to 5.0 mmol/l for subsequent diagnosis of GDM based on IADPSG criteria. We performed a Receiver Operating Characteristic analysis (Medcalc statistical software) to establish which FBG cut-point best correlated with elevated post-challenge BGLs.

### Ethics

The human research ethics committee of the Royal Brisbane and Women's Hospital, Queensland approved the study (HREC/15/QRBW/476). All data were fully anonymized before evaluation.

## Results

There were 26,683 OGTTs performed from 1 January 2015 to 30 June 2015 with incomplete data on 441 tests leaving 26,242 complete OGTTs available for further analysis. There were 3,946 (15.0%) patients with GDM. The diagnosis of GDM was established in 2,262 of these 3946 cases (57.3% of positive cases; 8.6% of the whole cohort) having an elevated FBG (+/- elevated post-challenge BGLs). The length of gestation was known in 16,358 of the whole cohort, 13,239 having the test performed between 24–32 weeks of gestation, matching the entry criterion of the HAPO study [1]. Finally, 11,450 patients had the OGTT between 24–28 weeks of gestation consistent with IADPSG recommendations [2]. The number of patients having a preceding 50gm glucose challenge test could be determined from data from PathQ and was less than 5% in the study period. In patients having an OGTT prior to 24 weeks, the relationship of FBG to the post-challenge BGLs was equivalent. As there was no differences in the overall rates of GDM in any of these subgroups compared with the whole cohort, data for the whole cohort were analysed.

   The relationship between fasting and post-challenge BGLs for FBG from 3.0 to 6.0mmol/l was assessed as less than 0.1% of patients had a FBG<3.0 mmol/l or >6.0mmol/l. A strong positive relationship between FBG and post-load BGLs was observed (Fig 1). At the FBG threshold (5.1 mmol/l) for the diagnosis of GDM the respective corresponding mean 1-hr and 2-hr BGLs were 8.3 ± 1.7mmol/l and 6.7 ± 1.4mmol/l, less than their corresponding diagnostic thresholds (2, 10). The overall numbers of patients, the point prevalence and the cumulative rates of elevated post-challenge BGLs at FBGs from 3.0 to 6.0mmol/l are shown (Table 1). A ROC analysis was performed to assess the optimal FBG for identifying patients having elevated

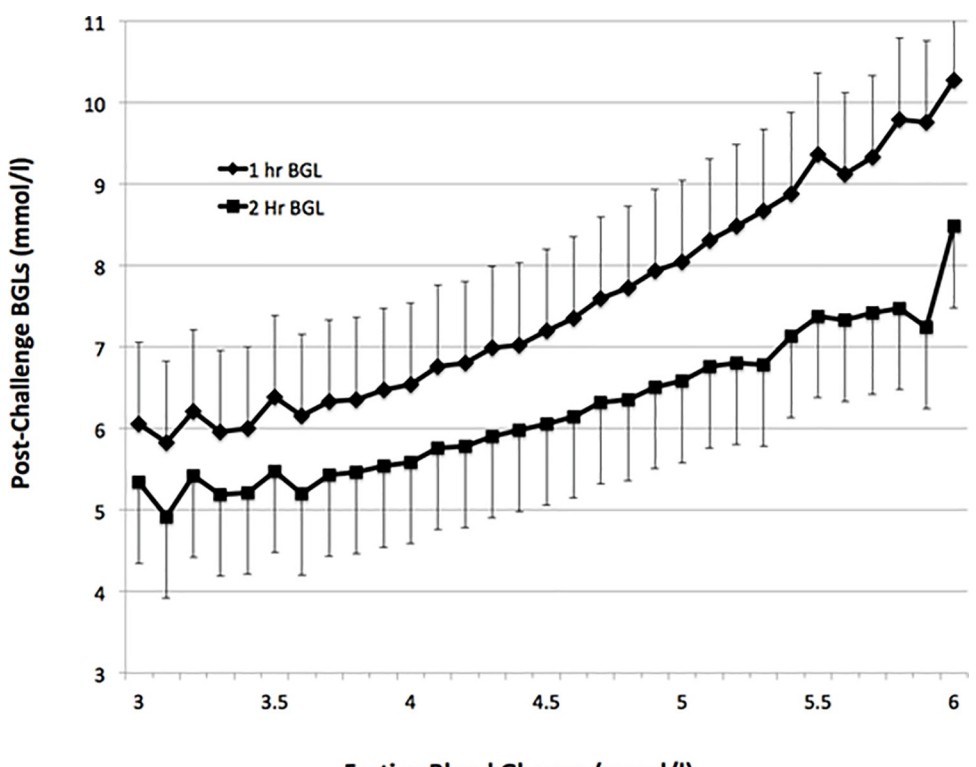

**Fig 1. Relationship between FBG and post-load BGLs.** Shown are the post-challenge BGL values (mean; standard deviation) on an oral glucose tolerance test in 26,242 patients between January 1, 2015 and June 30, 2015 from the three major laboratories in Queensland, Australia for FBGs between 3.0 and 6.0mmol/l. Less than 0.1% of patients had FBG below or above these values.

**Table 1. Analysis of screening FBG levels on rates of elevated post-challenge BGLs.**

| A. | B. | C. | D. | E. |
|---|---|---|---|---|
| FBG | Patients | Point estimate elevated post-load BGLs | Cumulative Number of patients | Cumulative number elevated post-load BGLs |
| mmol/l | n | n (%) | n (%) | n (%) |
| 4.0 | 1827 | 59 (3.2) | 5673 (21.6) | 184 (0.7) |
| 4.1 | 2151 | 110 (5.1) | 7824 (29.8) | 294 (1.1) |
| 4.2 | 2518 | 127 (5.1) | 10,342 (39.4) | 421 (1.6) |
| 4.3 | 2466 | 143 (5.8) | 12,808 (48.8) | 564 (2.1) |
| 4.4 | 2477 | 176 (7.1) | 15,285 (58.2) | 740 (2.8) |
| 4.5 | 2129 | 148 (7.0) | 17,414 (66.4) | 888 (3.4) |
| 4.6 | 1927 | 166 (8.6) | 19,341 (73.7) | 1,054 (4.0) |
| 4.7 | 1629 | 180 (11.1) | 20,970 (79.9) | 1,234 (4.7) |
| 4.8 | 1283 | 171 (13.3) | 22,253 (84.8) | 1,405 (5.4) |
| 4.9 | 1003 | 155 (15.5) | 23,256 (88.6) | 1,560 (5.9) |
| 5.0 | 724 | 124 (17.1) | 23,980 (91.4) | 1,684 (6.4) |

The table shows the number of patients having complete OGTTs (column B) at FBG levels from 4.0 to 5.0mmol/l (column A) for the 26,242 patients tested between 1 January 2015 and 30 June 2015. Column C documents the point prevalence of elevated post-load BGLs at each individual FBG level. Column D is the number of patients having a FBG equal to or less than stated FBG. Column E is the cumulative number of patients with a FBG less than or equal to the stated fasting blood glucose level having elevated post-load BGLs.

n = total number of cases; % = percentage of the total study cohort.

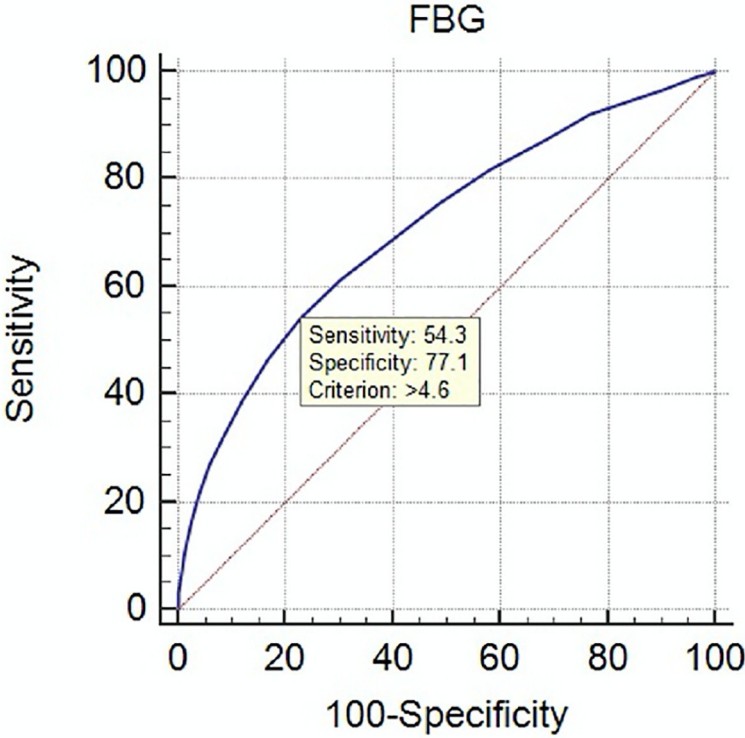

**Fig 2. ROC analysis of FBG and elevated post-load BGLs.** Shown is the ROC analysis (MedCalc statistical software) for the diagnosis of GDM based on elevated 1-hr and/or 2-hr BGLs on 75 gm OGTT for FBGs from 3.0 to 6.0 mmol/l. At FBG above 4.6 mmol/l, the sensitivity was 54.3% and specificity of 77.1%.

post-load BGLs (Fig 2). This suggested a FBG above 4.6mmol/l was the optimal predictor for elevated post-load BGLs with a sensitivity 54.3% and specificity 77.1%. There were 19,321 (73.7%) of the whole cohort having a FBG < 4.7mmol. Of these patients, only 1054 (4.1%) had any elevation of post-challenge BGLs, less than 1/3rd of the overall rate of GDM. There were 4,638 (17.7%) of the whole cohort having FBGs from 4.7 to 5.0mmol/l who would need further evaluation of their glucose tolerance. There were a further 630 cases of GDM (2.4% of all subjects) having elevated post-load BGLs in this subgroup having intermediate FBG levels. Of these cases, 363 of the 630 (1.4% of all subjects) had an elevated 2-hr post-load BGL.

Thus, based on an FBG < 4.7mmo/l, 19 of every 20 patients will be correctly diagnosed with GDM, advised to have further testing or reassured that they have a reduced risk of elevated post-challenge BGLs. The need for alternative strategies to assess glucose intolerance would be avoided in more than 4 out of 5 women. This represents a large reduction in time spent at collection facilities, minimising their exposure to a higher risk environment.

## Discussion

Our study reflects contemporary clinical practice across the state of Queensland, Australia. These results provide practical information to clinicians concerning the interpretation of the FBG and the risk of having elevated post-load BGLs. This should enable a discussion between a pregnant woman and her health practitioners concerning the need for a POGTT. Individual clinicians or units may choose to have alternative FBG threshold levels for recommending further evaluation of glucose tolerance in pregnancy based on these data, although the ROC

analysis suggested that a FBG > 4.6 mmol/l had the best sensitivity and specificity for having elevated post-load glucose levels, admittedly without high levels of specificity. The role of the FBG as a first step for the exclusion or diagnosis has been suggested by other investigators who found a similar BGL threshold on ROC analysis [11]. A lower level has been suggested from studies performed in Chinese populations [12, 13]. Others have recommended lower FBG thresholds [14–16] to reduce the need to perform POGTTs, referencing early outcome data reported from HAPO [2].

If used, patients with indeterminate FBGs from 4.7 mmol/l to 5.0 mmol/l will require further evaluation to define their glucose tolerance, which would typically be a POGTT. In the current environment, individual health practitioners, health centres or juristictions may use alternative strategies in this group of patients including an HbA1c, continuous glucose monitoring, home blood glucose monitoring or a subsequent 2-hr post-load BGL, as evidenced by the United Kingdom and Canadian recommendations during the COVID pandemic [4]. Alternative strategies to assess glucose tolerance in pregnancy have been previously evaluated. An HbA1c has not proved that useful for the diagnosis of GDM as opposed to overt diabetes [6, 17]. This was demonstrated in the analysis of the UK recommendations for diagnosis of GDM during the COVID-19 pandemic where a diagnostic strategy based in part on an HbA1c > 5.7% (39 mmol/mol) resulted in 81% of cases fulfilling IADPSG criteria being missed [4]. Home glucose monitoring using capillary glucose monitoring or continuous interstitial fluid glucose monitoring can be undertaken in the situation, for a week or two, or perhaps for the remainder of the pregnancy [18, 19]. Glucose monitoring strategies are more labour intensive, requiring education by the clinician, midwife or diabetes educator and incur significant expense, given the cost of monitoring strips, continuous glucose sensors, meters or reading devices and subsequent consultations for their interpretation. These strategies do not adequately exclude or establish the diagnosis of GDM if "normal" because their interpretation is not based on evidence-based thresholds for diagnosis. Rather, their interpretation is based on threshold BGLs for therapeutic intervention [18, 19]. The need for post-partum screening for the development of type 2 diabetes mellitus in these patients is unknown.

The findings of our analysis have been validated in a separate population group as a result of the need to reduce the risk of pregnant women being exposed to the Corona virus. Based on our data, an initial screening FBG at 24–28 weeks of pregnancy was recommended as an alternative strategy for diagnosing GDM in Queensland [20] with women having a FBG below 4.7mmol/l not requiring further evaluation. This strategy was subsequently recommended by the Australian Diabetes Society, ADIPS, Australian Diabetes Educators Association and Diabetes Australia [21]. The potential impact of this and other strategies has subsequently been analysed in a group of 5974 subjects from 5 HAPO research sites across several countries— Australia, Canada, England, Singapore and USA [4]. Based on the IADPSG criteria, an overall rate of diabetes of 17% was observed, similar to our rate of 15.0%. Using the Australian COVID-19 diagnostic criteria, 12.7% of subjects were diagnosed with GDM with 4.3% "missed", having FBGs less than 4.7mmol/l but having elevated post-load BGLs. This rate of missed diagnoses of GDM is very similar to our rate (4.0%).

The major potential problem of considering a FBG < 4.7 mmol/l as being "normal" is that 4.0% of these cases in our study and 4.3% in the subgroup of HAPO have elevated post-load BGLs and are not identified with GDM. The concern is whether these pregnancies are associated with higher rates of pregnancy-associated complications. We cannot assess this as we have no outcome data. However, data from HAPO provides reassurance that this is not the case. Firstly, in our analysis 70% of subjects had FBG < 4.5 mmol/l reported to be associated with a low risk of some adverse outcomes in the original HAPO study report, irrespective of their post-challenge glucose levels [2]. Secondly, an analysis of 6128 subjects from 5 HAPO

centres assessed several pregnancy-related outcomes in patients whose FBG was above or below the 75th centile (FBG < 4.6 mmol/l). This did not show any difference in adverse outcomes these women compared with women not diagnosed with GDM [22].

Finally, the recent analysis of the impact of the different the COVID-19 GDM diagnostic strategies reported the rates of adverse pregnancy outcomes [4]. The rates of adverse pregnancy outcomes in this missed group were compared to women not diagnosed with GDM by either the IADPSG or the Australian COVID-19 strategy. There was no increased rate of any adverse pregnancy in the "missed group" (Table 2). There was no difference in the rate of neonatal hypoglycaemia observed in the "missed GDM group". Although there was a non-significant increase in the number of cases of LGA, neonatal adiposity and neonatal hyperinsulinaemia, there was a lower number of subjects requiring a Caesarian section. Interestingly, a lower number of cases of pregnancy-associated hypertension was also observed. Thus, there was no evidence that a screening strategy based on a ROC-validated FBG less than 4.7mmol/l resulted in missing cases of GDM having an excess of important adverse pregnancy outcomes.

Our study has several strengths. The birth rate in Queensland during this period was estimated to be 30,829 based on the number of babies enrolled in the Queensland Newborn Screening Program. Therefore, our study captured the POGTT data of nearly 85% of all live births. The OGTT data is derived from tests performed by both private and public pathology providers likely to include all socio-economic, ethnic and geographical groups. At the last census in 2011, 20% of all Queenslanders were born overseas coming from 220 counties speaking over 220 languages [23]. Over 80% of OGTTs were tested between 24–32 weeks consistent with HAPO [1] and their results were no different to the whole cohort. A minimal number of patients had prior glucose challenge tests eliminating any significant bias by preferentially selecting subjects with elevated post-challenge BGLs. The similarity of our rates of glucose tolerance in pregnancy to the HAPO subgroup analysis validates our observations.

In conclusion, this analysis suggests that a 2-step diagnostic pathway based on an initial FBG can be used to identify women having either "normal" glucose tolerance associated with a low risk in pregnancy-associated adverse outcomes, or women with GDM. It reduces the need to perform POGTTs in the current environment by approximately 80%. These women with indeterminant FBGs will need to consider having a POGTT or undertake alternative strategies to assess glucose tolerance. Importantly, adopting this 2-step approach still enables the

**Table 2. Rates of adverse pregnancy outcomes in subgroup of women in HAPO (adapted from data of McIntyre et al; Ref 4).**

| Adverse Pregnancy Outcome | Missed GDM | | Non-GDM | | Excess Cases Missed | |
|---|---|---|---|---|---|---|
| | C/Tc | % | C/Tc | % | C | % |
| Pregnancy-related hypertension | 23/242 | 9.5 | 709/4856 | 14.6 | -12.4 | -5.1 |
| Preterm | 18/253 | 7.1 | 264/4981 | 5.3 | 4.6 | 1.8 |
| Large-for-gestational age | 28/253 | 11.1 | 398/4975 | 8.0 | 7.8 | 3.1 |
| Primary Cesarean section | 33/216 | 15.3 | 758/4407 | 17.2 | -4.1 | -1.9 |
| Neonatal hyperinsulinemia | 27/229 | 11.8 | 311/4380 | 7.1 | 10.8 | 4.7 |
| Neonatal hypoglycemia | 33/194 | 17.0 | 634/3686 | 17.2 | 0.4 | -0.2 |
| Neonatal adiposity | 20/204 | 9.8 | 331/4037 | 8.2 | 3.3 | 1.6 |

C = Number of cases in the original publication.

Tc = calculated number of subjects having data, based on the percentage of patients reported to have the adverse outcome (the absolute number of cases not being reported in the publication; Tc = [Cx100]/%).

"Excess cases missed" is the difference in number of adverse outcomes associated with elevated glucose levels observed in women with a missed diagnosis of GDM compared with the expected rate of these complications, based on the rate observed in women not diagnosed with GDM (non-GDM).

effective identification of cases of GDM at increased risk of adverse pregnancy outcomes [4], including those at highest risk having elevated fasting and post-load BGLs [24]. It negates the need to do an POGTT if an FBG has been performed for other reasons as our study provides data to make an adequate risk assessment in this circumstance. It is a simple, in-expensive, reliable and readily understood measurement of glucose, better tolerated than the POGTT that can have a role in the assessment of glucose intolerance in pregnancy in the current COVID-19 environment, in health care environments with limited resources and potentially in other clinical circumstances eg patients post bariatric surgery. These date when considered together with the outcome data from a subgroup of HAPO suggests that this two step-diagnostic approach could be considered as a more efficient and cost-effective strategy for the diagnosis of GDM. It would significantly reduce the need for a POGGT and avoid labelling 25% of women with a diagnosis of GDM whose pregnancies are not associated with increased rates of adverse outcomes, avoiding providing un-necessary additional resources for education, blood glucose monitoring or therapeutic intervention.

## Supporting information

**S1 File.**
(XLSX)

## Author Contributions

**Conceptualization:** Michael d'Emden.

**Data curation:** Michael d'Emden, Donald McLeod, Jacobus Ungerer, Charles Appleton, David Kanowski.

**Formal analysis:** Michael d'Emden, Donald McLeod.

**Investigation:** Michael d'Emden.

**Methodology:** Michael d'Emden.

**Project administration:** Michael d'Emden.

**Supervision:** Michael d'Emden.

**Validation:** Jacobus Ungerer.

**Writing – original draft:** Michael d'Emden.

**Writing – review & editing:** Michael d'Emden, Donald McLeod, Jacobus Ungerer, Charles Appleton, David Kanowski.

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
