## [Decision Letter · Decision Letter 0]

17 Sep 2020

PONE-D-20-26829

Development of a fasting blood glucose-based strategy to diagnose women with gestational diabetes mellitus at increased risk of adverse outcomes in a COVID-19 environment.

PLOS ONE

Dear Dr. d'Emden,

Thank you for submitting your manuscript to PLOS ONE. After careful consideration, we feel that it has merit but does not fully meet PLOS ONE’s publication criteria as it currently stands. Therefore, we invite you to submit a revised version of the manuscript that addresses the points raised during the review process.

An expert in the field handled your manuscript, and we are very thankful for their time and efforts. Although interest was found in your study, there are some comments that arose that require your attention. Please address ALL of the reviewer's comments in your revised manuscript.

We look forward to receiving your revised manuscript.

Kind regards,

Frank T. Spradley

Academic Editor

PLOS ONE

2. In your ethics statement in the Methods section and in the online submission form, please clarify whether all data were fully anonymized before you accessed them and/or whether the IRB or ethics committee waived the requirement for informed consent.

3. Please provide a table of baseline characteristics for the subjects in the study.

4. Please include the date(s) on which you accessed the databases or records to obtain the data used in your study.

7. Your ethics statement should only appear in the Methods section of your manuscript. If your ethics statement is written in any section besides the Methods, please move it to the Methods section and delete it from any other section. Please ensure that your ethics statement is included in your manuscript, as the ethics statement entered into the online submission form will not be published alongside your manuscript

8.Thank you for stating the following in the Acknowledgments Section of your manuscript:

[DM is supported by an NHMRC Early Career Fellowship.]

 [The author(s) received no specific funding for this work.]

Reviewers' comments:

Reviewer's Responses to Questions

**Comments to the Author**

1. Is the manuscript technically sound, and do the data support the conclusions?

Reviewer #1: Yes

2. Has the statistical analysis been performed appropriately and rigorously? 

Reviewer #1: Yes

3. Have the authors made all data underlying the findings in their manuscript fully available?

Reviewer #1: Yes

4. Is the manuscript presented in an intelligible fashion and written in standard English?

Reviewer #1: Yes

5. Review Comments to the Author

Reviewer #1: The manuscript is technically sound and the analysis of the de-identified data support the conclusions.The ROC analysis helped to justify the preferred use of FPG over OGTT during this pandemic.

Also, the figures where explicit enough,though other additional data not included should be added to make the research work more robust and clearer. The use of English was simple and unambiguous,however there were some minor mistakes identified which include the following

i. Line 5 Abstract: POGTT in the diagnosis of GDM NOT POGTT to diagnose GDM

ii. Line 10: high risk NOT higher risk

iii Page 17, line 18: Our study was a real life study---- is not necessary since the study is centered on de identified data.

iv. Page 25, Reference 2 and 14 not well written

Delibrate more on this statement in your discussion as this is a rather controversial statement you made"An HbA1c has not proved that useful for the diagnosis of GDM as opposed to overt diabetes [6, 17]".

6. PLOS authors have the option to publish the peer review history of their article (what does this mean?). If published, this will include your full peer review and any attached files.

Reviewer #1: No

---

## [Author Response · Author response to Decision Letter 0]

11 Oct 2020

Point 1. 

I believe our submitted revised manuscript meets the style requirements of your Journal. 

Point 2 

I have clarified whether the data was fully anonymised by including the following in the Ethics statement:- 

The human research ethics committee of the Royal Brisbane and Women’s Hospital, Queensland approved the study (HREC/15/QRBW/476). All data were fully anonymized before evaluation. 

Point 3 

As the data was obtained from biochemistry laboratories, there was minimal baseline characteristics so we really cannot provide a table of baseline characteristics for the subjects in our study. I have made a slight amendment in the methods statement to clarify this:- 

Minimal baseline (date of birth, age, date of test, length of gestation) and no outcome or management data were available.

Point 4

I have added the following to the methods section to state when the data was accessed. It was obtained at different times from the three laboratories. The methods section now includes:- 

We evaluated the de-identified data of all patients having POGTTs (data available at 0, 1 and 2-hrs) at the three largest pathology laboratories (Queensland Medical Laboratory [QML], Sullivan Nicolaides Pathology [SNP], and Pathology Queensland (PathQ]) in the state of Queensland, Australia from January 1, 2015 to June 30, 2015 which was accessed in late 2015. 

Point 5. 

I will upload an excel file which contains all the de-identified data analysed in this study 

Point 6.

I have removed the phrase “data not shown” as per your suggestion as it does not add to the manuscript. The POGTTs having prior 50gm glucose screening tests are identified in the uploaded de-identified dataset. 

Point 7. 

I have insured that the Ethics Statement only appears in the Methods section. 

Point 8. 

I have removed “DM is supported by an NHMRC Early Career Fellowship” from acknowledgments and have not added it to the Funding Statement. This was part of his salary and not specifically related to the conduct of this research. I don’t believe it should be considered as “funding”.

As to the specific comments of the reviewer:-

We have modified line 5 of the abstract so that is now reads POGTT for the diagnosis of gestational diabetes mellitus (GDM). We felt “for the diagnosis” was better wording than “in the diagnosis”. We trust you agree. 

ii) Line 10. “Higher risk” has been changed to “high risk”.

iii) We have removed “real life study” accepting the reviewers comments. iv) We have corrected references 2 and 14 

We have included the analysis of the UK Covid -19 diagnostic approach to GDM which is based, in part, on an HbA1c of 5.7% that results in 81% of cases of GDM being missed. We think that this clearly illustrates that an HbA1c is not a good screening test for GDM. It now reads:- 

An HbA1c has not proved that useful for the diagnosis of GDM as opposed to overt diabetes [6, 17]. This was demonstrated in the analysis of the UK recommendations for diagnosis of GDM during the COVID-19 pandemic where a diagnostic strategy based in part on an HbA1c > 5.7% (39 mmol/mol) resulted in 81% of cases fulfilling IADPSG criteria being missed [4]. 

I have made some very minor changes to the text which will have no bearing on meaning or content and these are visible on the “marked up version”. I have re-checked all the references and believe they are in the correct format for your Journal, ensuring that the first 6 authors are listed on several. I have rechecked the links to on-line publications and needed to change one link due to it being moved to another site.

---

## [Decision Letter · Decision Letter 1]

18 Nov 2020

Development of a fasting blood glucose-based strategy to diagnose women with gestational diabetes mellitus at increased risk of adverse outcomes in a COVID-19 environment.

PONE-D-20-26829R1

Dear Dr. d'Emden,

We’re pleased to inform you that your manuscript has been judged scientifically suitable for publication and will be formally accepted for publication once it meets all outstanding technical requirements.

Kind regards,

Frank T. Spradley

Academic Editor

PLOS ONE

Reviewers' comments:

Reviewer's Responses to Questions

**Comments to the Author**

1. If the authors have adequately addressed your comments raised in a previous round of review and you feel that this manuscript is now acceptable for publication, you may indicate that here to bypass the “Comments to the Author” section, enter your conflict of interest statement in the “Confidential to Editor” section, and submit your "Accept" recommendation.

Reviewer #1: All comments have been addressed

2. Is the manuscript technically sound, and do the data support the conclusions?

Reviewer #1: Yes

3. Has the statistical analysis been performed appropriately and rigorously? 

Reviewer #1: Yes

4. Have the authors made all data underlying the findings in their manuscript fully available?

Reviewer #1: Yes

5. Is the manuscript presented in an intelligible fashion and written in standard English?

Reviewer #1: Yes

6. Review Comments to the Author

Reviewer #1: All issues have been addressed by the authors. Satisfactory and fit for publication.All data well written

7. PLOS authors have the option to publish the peer review history of their article (what does this mean?). If published, this will include your full peer review and any attached files.

Reviewer #1: No

---

## [Editor Report · Acceptance letter]

23 Nov 2020

PONE-D-20-26829R1 

Development of a fasting blood glucose-based strategy to diagnose women with gestational diabetes mellitus at increased risk of adverse outcomes in a COVID-19 environment. 

Dear Dr. d'Emden:

I'm pleased to inform you that your manuscript has been deemed suitable for publication in PLOS ONE. Congratulations! Your manuscript is now with our production department. 

Kind regards, 

on behalf of

Dr. Frank T. Spradley 

Academic Editor

PLOS ONE